# Influence of Feedstock in the Formation Mechanism of Cold-Sprayed Copper Coatings

Jui-Ting Liang [1,2], Shao-Fu Chang [1], Cheng-Han Wu [1], Shih-Hsun Chen [1,2,*], Che-Wei Tsai [2,3], Kuei-Chung Cheng [4] and Kim Hsu [5]

1    Department of Mechanical Engineering, National Yang Ming Chiao Tung University, Hsinchu 30010, Taiwan; ray840725@gmail.com (J.-T.L.); shaofu78@gmail.com (S.-F.C.); james8th0805@gmail.com (C.-H.W.)
2    High Entropy Materials Center, National Tsing Hua University, Hsinchu 30013, Taiwan; chewei@mx.nthu.edu.tw
3    Department of Materials Science and Engineering, National Tsing Hua University, Hsinchu 30013, Taiwan
4    Plus Metal Tech Ltd., Tainan 71247, Taiwan; maxckc123@gmail.com
5    Integrated Service Technology, Hsinchu 300094, Taiwan; kimhsu@istgroup.com
*    Correspondence: brucechen@nycu.edu.tw; Tel.: +886-3-5712121-55117

**Abstract:** The aim of this article is to investigate the characterizations and formation mechanisms of cold-sprayed coatings using gas-atomized and electrolytic powders. The study highlights the importance of reaching the particles' critical velocity for optimal deposition. The main findings reveal that the morphology and stacking conditions of the coatings have a significant impact on their mechanical properties. Coatings made with gas-atomized powders exhibited noticeable pores and higher plastic deformation, while electrolytic powder coatings had greater density and smoother interfaces with the substrate. Adhesion strength relied on the physical bonding resulting from the plastic deformation energy between the spraying powders and the substrate. Gas-atomized powders showed higher adhesion compared to electrolytic powders, with dendritic powders resulting in lower adhesion due to dispersed impact force. The interaction between thermal and kinetic energy during the cold spraying process facilitated plastic deformation and particle deposition by softening and eroding the substrate surface. Insufficient plastic deformation with dendritic powders led to incomplete overlap, pore formation at the interface, and the persistence of the oxide layer along powder boundaries. Overall, these findings provide valuable insights into the influence of powder properties on coating morphology, adhesion strength, and overall performance, contributing to the understanding and optimization of cold spray processes.

**Keywords:** cold spray; adhesion strength; formation mechanism; gas-atomized; electrolysis

## 1. Introduction

With the rapid advancement of internet technology in recent years [1–4], the importance of computational power in devices has significantly increased. This has led to a growing demand for high-density integrated circuits and more advanced functionalities in active components compared to the past. Moreover, the global pandemic that unfolded over the last three years has generated a new work-from-home trend, resulting in a sudden surge in the need for electronic products within a short span of time. Consequently, the electronic industry has experienced another remarkable growth spurt. Among the various electronic products, printed circuit boards (PCBs) have emerged as highly sought-after items. PCBs serve as essential substrates used in nearly all electronic devices, facilitating the connection between individual electronic components. They typically consist of a layered structure, with conductive and insulating layers sandwiched on a non-conductive substrate. These well-designed copper circuits feature soldered conductive pads, enabling effective communication between the electrical components.

The establishment of copper conductive layers on PCBs typically involves electroplating or bonding rolled and annealed copper foil. Once the copper layer is in place, a series of chemical processes are employed to create the desired wire patterns. However, concerns regarding wastewater treatment have prompted the search for alternative solutions to mitigate the environmental and health risks associated with copper electroplating [5]. In the 1980s, researchers from the Siberia Branch of the former USSR Academy of Sciences made a significant discovery during a cavity experiment. They found that metallic powders could be effectively adhered to target areas, forming a continuous and dense coating when propelled by supersonic airflow. This technology caught the attention of researchers in the United States and Germany who also began exploring the use of gas-powered spraying of metallic powders [6–9]. Referred to as thermal spray processes, these methods involve the use of a heat source, typically a flame or plasma, to soften or melt the feedstock. High-pressure working gases provide the necessary energy to propel the molten material onto the target surface. However, the high temperature during spraying can cause feedstock oxidation, leading to deviations from the desired coating properties. Consequently, it is crucial to explore non-fusion-based technologies that do not alter the properties of the deposited material during the process.

In the past decade, a deposition process called cold spraying has gained significant momentum across various fields. This technique involves the use of un-melted metal particles to create a metallic coating on the target surface [10–16]. Cold spraying stands out as a chemical-free process that eliminates the release of waste liquid and exhaust gases. It offers faster and more precise performance compared to traditional spraying technologies, while generating minimal dust pollution that is easy to manage. Unlike thermal spray processes [17–19], cold spray offers several advantages. Firstly, the working temperature in cold spraying is controlled below the melting point of the supplied powder. This prevents significant oxidation during the spraying process, ensuring that the initial phase constitution of the material is retained and minimizing deviations from the design. Secondly, the absence of boiling and melting in cold spray materials allows the coating to be formed without gas cavity, resulting in a denser film structure. Additionally, the safe and flameless working conditions of cold spray enable its application in a broader range of areas. Given these advantages, the objective of this study is to investigate the impact of feedstock type on the formation of copper coatings using the cold spray process.

Due to its numerous advantages and the increasing demands of the industry, the development of the cold spray process offers not only an environmentally friendly alternative but also opens up possibilities for the application of other active metals, such as Mg, Ti, and Al, in the field of spraying. In order to explore the potential of cold spray technology, various types of commercial copper powders were utilized instead of conventional copper foils. The study aimed to investigate the surface roughness, microstructure, deposition rate, and adhesion strength of the coatings formed using these different copper powders, with the ultimate goal of elucidating the underlying formation mechanism. This research not only provides an additional option for the industry but also paves the way for further advancements in the cold spray process and its application to a wider range of materials.

## 2. Materials and Methods

### 2.1. Characterizations of Cu Powder Feedstock

During the cold spraying process, particles are accelerated to supersonic speeds and directed towards the target area. Through plastic deformation, these particles form a solid bond and create a stacked coating. The performance of the coating depends on the properties of the feedstock, which undergoes heating and acceleration during the process. These properties ultimately determine the variations observed in the resulting products. While the energies governing the powders can be regulated by adjusting carrier gas conditions, such as temperature and ejection pressure, the range of control is limited due to the constraints of using inert gas as the medium. To gain a clearer understanding of the formation mechanism underlying this emerging deposition technology, the process

conditions were kept fixed to isolate the effects caused by the type of feedstock. For this study, five different commercial copper powders with a purity higher than 99.9% were selected to investigate their influence on the film properties. Among these powders, two were gas-atomized, while the other three were prepared using electrolysis, serving as the control variables for comparison.

The gas atomization process involved smelting copper bulk in an induction furnace under an argon environment. The resulting molten alloy was then poured into a gas nozzle, passing through an insulation tundish. Subsequently, high-pressure inert gas was introduced to disintegrate the molten alloy into droplets at a cooling rate of $10^5$ K/s. Due to solidification during flight, the gas-atomized powders exhibited a spherical shape and followed a normal distribution. By adjusting the smelting temperature and nozzle pressure, powders with a smaller average size could be produced. The morphology of the two atomized powders, labeled as A1 and A2 based on their size levels, was examined using SEM (JEOL, JSM7900F, Tokyo, Japan), as shown in Figure 1a,b. The size distribution of A1 powders was characterized by D50 and D90 values of 24.8 and 45.7 μm, respectively, while the D50 value for A2 was 64.6 μm, as detailed in Table 1.

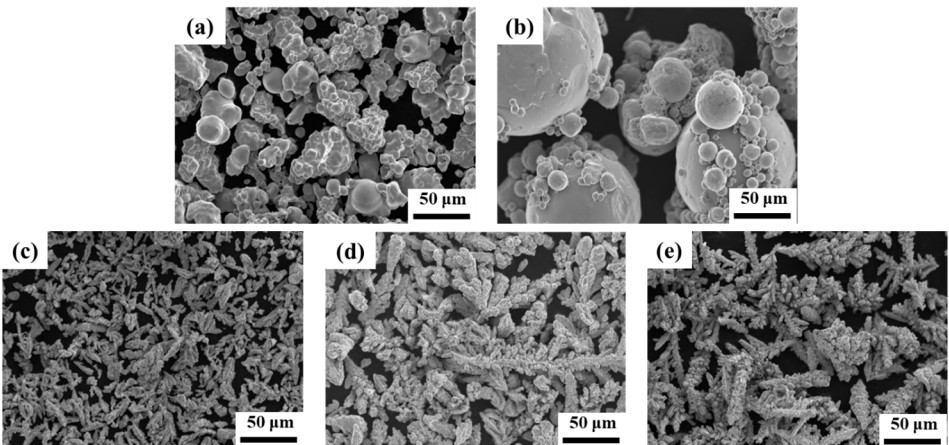

**Figure 1.** SEM images showing the appearance of gas-atomized powders (**a**) A1 and (**b**) A2, and the powders synthesized using electrolysis process (**c**) E1, (**d**) E2, and (**e**) E3.

**Table 1.** The classification and average particle sizes of five kinds of powder feedstock.

| Powder Name | | A1 | A2 | E1 | E2 | E3 |
|---|---|---|---|---|---|---|
| Manufacturing Process | | Gas atomization | | Electrolysis | | |
| Average Particle Size (μm) | (D50) | 24.8 | 64.6 | 20.1 | 28.6 | 36.9 |
| | (D90) | 45.7 | 137.0 | 50.1 | 66.8 | 84.2 |

Regarding the electrolytic powders, the reduction reaction was initiated by an electric current, leading to the direct deposition of copper ions from the sulfate electrolyte onto the cathode. At high current densities, both hydrogen and copper would simultaneously precipitate, resulting in the formation of fine and loosely packed crystalline powders. Unlike the wide distribution observed in the atomization process, the reduction of metal ions in electrolysis powders depends on various factors, including the electrolyte, current density, working temperature, and the nature of the cathode. As a result, there is a limited variety in the characteristics of the final electrolysis powders. The appearance of the three types of electrolysis powders, labeled as E1 to E3, displayed a dendritic structure. These structures are depicted in Figure 1c–e, respectively. Due to their irregular shapes and high aspect ratios, accurately determining their particle size distribution can be challenging, despite being prepared through different processes. Upon closer inspection of the images, E1 appeared to be the thinnest, while E2 exhibited a coarse shape with medium length. E3, on the other hand, possessed a thin and elongated structure. The particle distributions

(D50) of the three electrolysis powders were measured using a Laser Scattering Particle Size Analyzer (Malvern Mastersizer 3000, Malvern, UK) and are described in Table 1.

*2.2. Preparation of Cold-Sprayed Cu Coatings*

In the cold spraying process, the raw powder is injected onto the substrate at high velocity, leading to its interaction with the target surface and subsequent plastic deformation [20,21]. This process results in the formation of a stacked coating and a strong bond. To ensure effective adhesion of the cold-sprayed coatings, it is important to use a relatively soft substrate, allowing the injected particles to penetrate the surface layer and establish interlock bonding [1]. For this study, 6061 aluminum alloy was selected as the substrate, and the cold spraying was performed using a High-pressure Cold Spray Equipment (PLASMA Cold Spray PCS-1000, Saitama, Japan), and the spraying parameters are specified in Table 2. In this procedure, the kinetic energy of the powder plays a crucial role in the formation of the spraying layer. Nitrogen gas is commonly used as the working fluid in cold spraying due to cost considerations. The working temperature is determined by the melting point of the feedstock, while the spraying pressure is limited to ensure the safety of the pressure vessel. With the given settings, each powder was individually subjected to reciprocating spraying four times, and subsequently, the characteristics of the cold-sprayed coatings were evaluated.

**Table 2.** Parameters of Cu cold spraying process.

| Working Gas | Preheating Temperature | Spraying Pressure | Working Temperature | Relative Traversing Speed (m/min) | Pitch (mm) | Powder Feed Rate (g/min) | Pass |
|---|---|---|---|---|---|---|---|
| $N_2$ | 100 °C | 5 Mpa | 800 °C | 100 | 1.2 | 100 | 4 |

The microstructure of the powders and coatings was examined using optical microscopy (Olympus BX-51, Tokyo, Japan) and FE-SEM (JEOL, JSM7900F, Tokyo, Japan). Energy-Dispersive Spectroscopy (EDS) (Oxford Instruments AZtec, High Wycombe, UK) was employed to obtain elemental mappings and chemical compositions. XRD analysis was carried out using a D2 PHASER XE-T X-ray Powder Diffractometer (XRD, Bruker, Billerica, MA, USA). Finally, the adhesion strength, a key property of the coatings, was measured using a Universal Testing Machine (HT-9501, Hung Ta Instrument Co., Ltd., Taichung, Taiwan) according to ASTM C633 standard [22,23].

## 3. Results and Discussion

*3.1. Characterizations of Cold-Sprayed Coatings Constructed Using Gas-Atomized and Electrolytic Powders*

In the cold spray process, an inert gas at high pressure and temperature is used to transport the feedstock powder through a nozzle, resulting in its high-velocity impact on the substrate. Achieving optimal deposition requires the particles to reach their critical velocity before reaching the substrate. However, the critical velocity of the particles may vary based on different powder particle sizes and working parameters. To assess the quality of the coatings, examining the cross-section of the coatings is crucial. It has been mentioned in the literature that the morphology of the powder stacking in the coating can significantly influence the mechanical properties of the coating. Therefore, in this study, five groups of powders with different particle size distributions and surface morphologies were utilized as spraying feedstock. The stacking conditions of the coatings were compared by analyzing the cross-sectional microstructure, as depicted in Figure 2.

Given that the microstructure of the coatings formed with electrolytic powders did not exhibit significant differences, the A1, A2, and E1 coatings, which displayed noticeable distinctions, were primarily chosen for comparison. Figure 2a,b depict the A1 coating, characterized by a lamellar structure formed by fully deformed powders, while the A2 coatings exhibited more undeformed particles dispersed throughout the layered structure. The larger size of A2 powders allowed them to penetrate deeper into the aluminum alloy

substrate due to higher impact energy, but their bigger size hindered sufficient deformation. In Figure 2c, the E1 coating, deposited using electrolysis copper powders, displayed a dense and flat stacking pattern with smaller particles. Additionally, the interface between the substrate and deposit appeared relatively smooth with minimal interlocking deformation. The adhesion of spraying coatings primarily relies on the physical bonding resulting from the plastic deformation energy between the spraying powders and the substrate [1]. Consequently, the particle stacking in the coating prepared using gas atomization contained relatively noticeable pores, and the degree of plastic deformation of the particles was higher, while the coating made of dendritic powders exhibited greater density. It can be presumed that the dendritic structure enhances contact points during impact and disperses the impact force, resulting in a relatively smooth interface between the coating and the substrate. Although a smooth surface finish can be achieved using electrolysis powders, the quality of adhesion may be a concern and warrants further examination through pull-off tests [23].

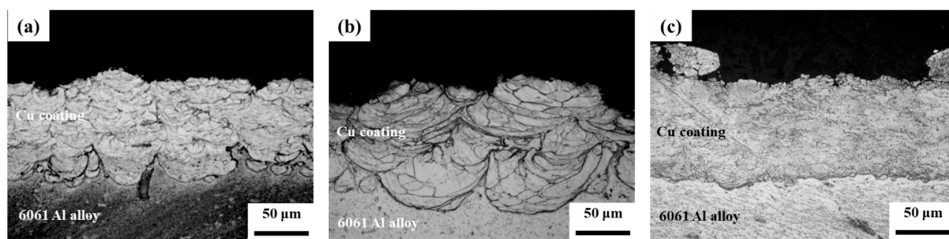

**Figure 2.** Cross-sectional views of the microstructure of the coatings constructed by (**a**) A1, (**b**) A2, and (**c**) E1 powders, respectively.

Prior to evaluating the performance of the coatings, it is important to verify the characteristics of the materials to ensure there are no deviations during the spraying process. Comparing the original powders with the standard lattice database of copper (JCPDS card No. 04-0836, Cu), both in terms of morphology and particle size differences, all five powder samples exhibit a pure copper phase with a face-centered cubic (FCC) structure. Furthermore, no significant oxidation signals are observed, as depicted in Figure 3a. After cold spraying, the coatings display the same pure copper phase with a tendency towards the FCC structure, indicating that the coating retains its initial phase and crystallinity following the cold spraying process. During the high-pressure cold spraying process, the spraying powder is softened at a temperature below its melting point and then propelled onto the substrate at ultrasonic speed. As a result of the interaction between thermal energy and kinetic energy, the particles and the substrate undergo substantial plastic deformation. The outer oxide layers of the powders and the substrate are eroded by the undeposited and rebounded particles, resulting in similar properties between the coating and the copper powders. Therefore, the coating exhibits an identical elemental composition and phase composition to the as-prepared powders, as illustrated in Figure 3b.

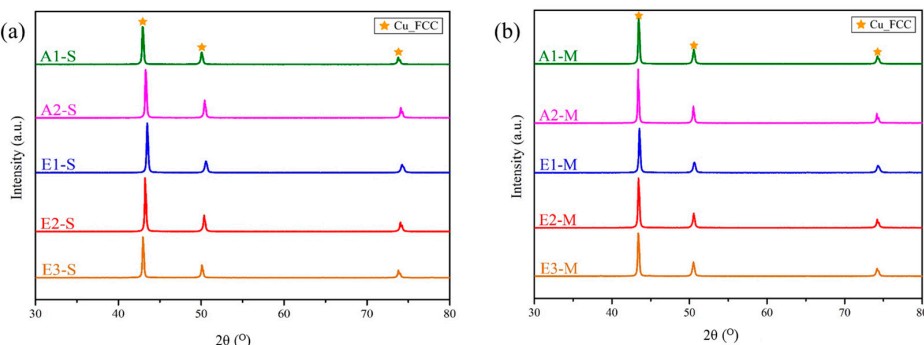

**Figure 3.** (**a**) XRD spectra of various Cu powders and (**b**) those of cold-sprayed coatings constructed with the same powders.

*3.2. Relationship between the Adhesion Strength of Powder and the Cold-Sprayed Coating*

In the cold spraying process, the powder feedstock is propelled by high-pressure gas with a temperature below the melting point. It is then sprayed onto the substrate, creating a stable bond through plastic deformation. The adhesion strength of the coating depends on the interactions between the spraying powders and the substrate, which can be influenced by various parameters. To investigate the impact of feedstock on the coating structure, five types of copper powders were used, and a pull-off adhesion test was conducted.

The bond strength between the coating and the substrate was evaluated using the ASTM C633 standard. The pull-off test was performed using a universal material tester with a fixed rate of vertical tension, as depicted in Figure 4a. Prior to testing, the coating was applied to a standard substrate specimen with a diameter of 25.4 mm, and epoxy adhesive was used to bond the loading specimen with the coated substrate specimen. Once the bonding process was complete, the sample combination was affixed to the universal material tester, and an axial force was applied to the specimen. The test concluded when the sample broke, and the maximum tensile force was recorded and converted into adhesion strength during the pull-off process. Figure 4b illustrates different examples that indicate the degree of bonding. Case 1 implies that no copper deposit remained on the substrate side, suggesting that the bonding strength of the adhesive is higher than that of the coating. Conversely, case 5 indicates that the bonding strength of the spraying coating is higher than the adhesive, resulting in fracture within the adhesive. Cases 2 to 4 demonstrate the progressive development of the coating's adhesion strength in a positive direction.

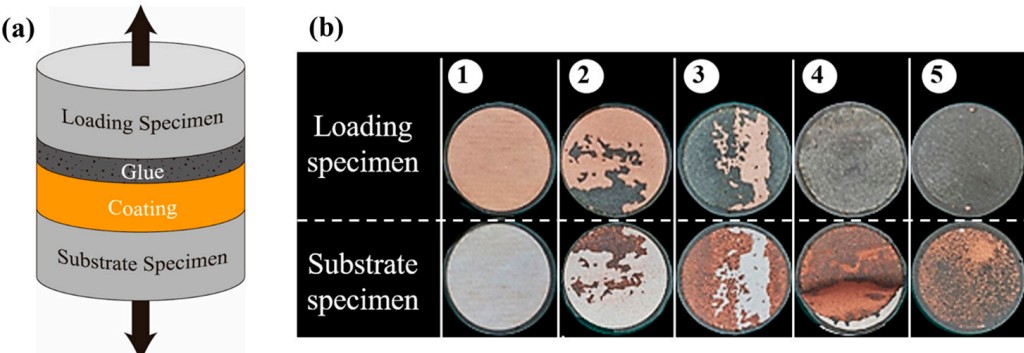

**Figure 4.** (**a**) Schematic diagram of the sample for pull-off adhesion test. (**b**) Examples of varying degrees of peeling coating. From 1 to 5, showing the increase in adhesion.

The results of the adhesion testing for the five spraying samples are presented in Figure 5. Overall, the gas-atomized powders exhibit higher adhesion compared to the electrolysis powders. The coatings prepared with spherical A1 and A2 powders demonstrate a coating adhesion greater than 70 MPa, despite A1 powder having a smaller particle size than the dendritic E2 and E3 powders. This suggests that the powder morphology has a stronger influence on coating adhesion than the powder size. The analysis of microstructure reveals that the variation in powder morphology primarily causes differences in powder impact force or energy. The dendritic powder, due to its irregular morphology, disperses the impact force on the substrate, resulting in lower coating adhesion. For the dendritic powders, the coating adhesion tends to increase with larger particle sizes, further emphasizing the significance of particle mass in adhesion. As depicted in Figure 6, the E3 powder with the highest average particle size exhibits the least peeling-off of coating, while the E1 coating presents a less favorable bonding situation.

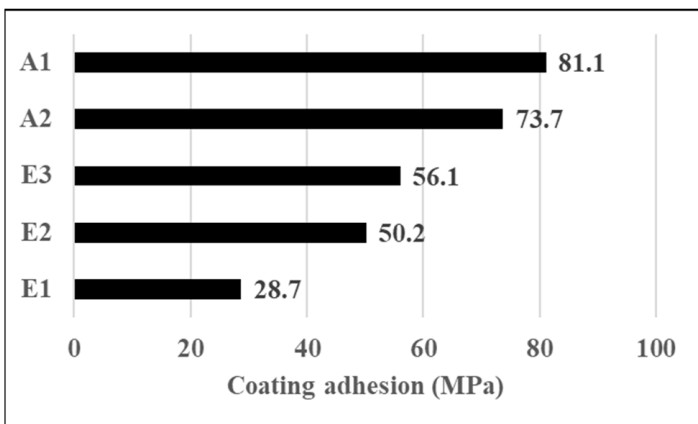

**Figure 5.** The comparison in the adhesion strengths of various cold-sprayed Cu coatings.

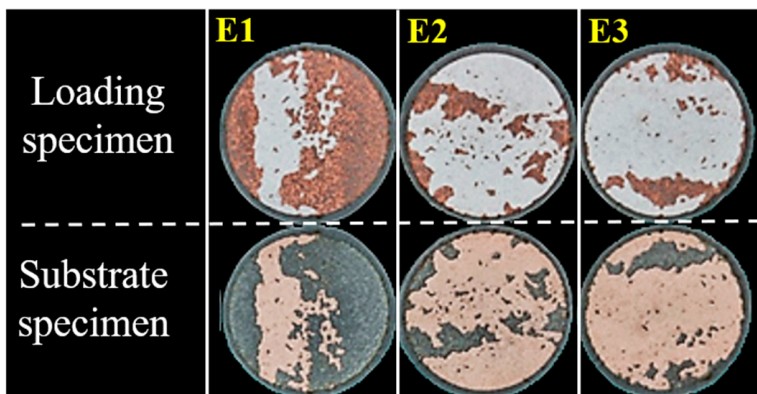

**Figure 6.** Observation in the fractured surfaces of pull-off testing specimens prepared using different electrolytic powders.

*3.3. The Formation Mechanism of Coating in Cold Spray Process*

Taking a closer look at the stacking of cold-sprayed Cu coatings, Figure 7 provides a cross-sectional view of coatings prepared with A1 and E3 powders. To ensure clear and concise observations, only one pass spraying was conducted on the aluminum alloy substrates. In reference to the stacking of spherical powders, it is evident that the initial particles forcefully penetrate the substrate surface and interconnect with each other, while the subsequent particles adhere firmly to the existing layer with substantial plastic deformation. In Figure 7a,b, the substrate surface undergoes deformation due to the impact of the injected particles, and the stacking morphology exhibits significant distortion caused by the high impact energy. On the other hand, the loose nature of electrolysis powders results in the cold-sprayed coating not penetrating the substrate, and the contour of dendritic particles can still be observed within the coating structure shown in Figure 7c,d, indicating a lower impact strength.

Based on the aforementioned results, a proposed formation mechanism for coatings prepared using the cold spray process is as follows: Firstly, the inert gas is heated to a specific temperature, and the powder feedstock is introduced into the system chamber filled with the working gas. It is then accelerated and transported through the nozzle to the substrate, as depicted in Figure 8. Upon impact with the substrate, the kinetic energy is converted into heat energy, which softens and erodes the surface, removing a portion of the oxides. Ideally, a favorable bonding scenario in cold spraying is achieved when both the substrate and particles undergo sufficient plastic deformation, resulting in particle deposition on the substrate in a crater-like shape, as illustrated in Figure 8c. When there is a significant height difference at the substrate interface, the coating tends to exhibit higher adhesion strength.

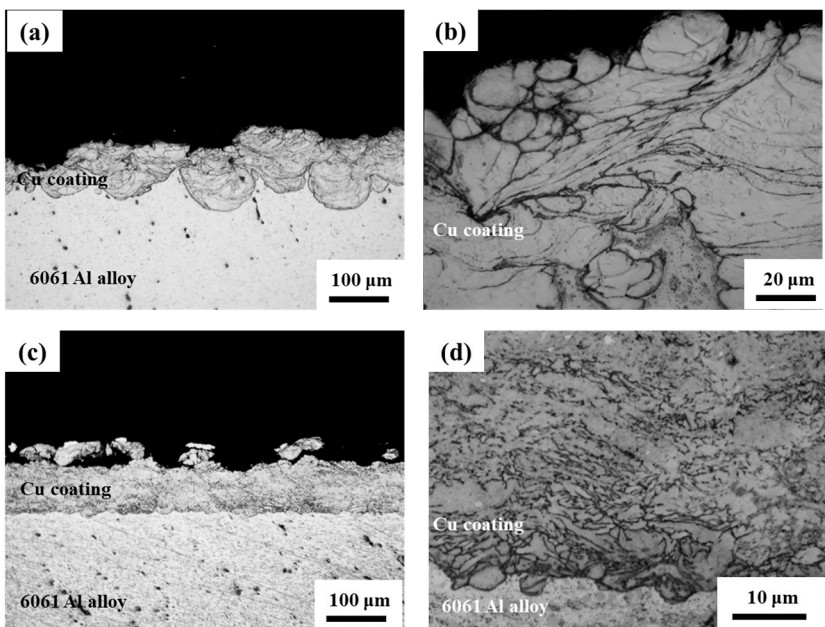

**Figure 7.** Cross-sectional views of the microstructure of the one-time spray coatings constructed by (**a**,**b**) A1 and (**c**,**d**) E3 powders, respectively.

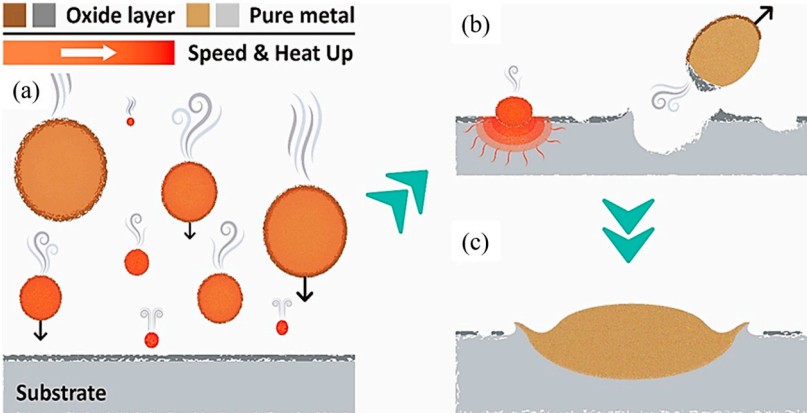

**Figure 8.** Schematic diagrams illustrating the powder deposition mechanism in the cold spray process: (**a**) The powders undergo acceleration and varying degrees of softening influenced by their particle sizes. (**b**) The spraying powders either erode or adhere to the substrate surface. (**c**) The softened powders undergo plastic deformation and are deposited on the substrate, forming a crater-shaped structure.

Furthermore, apart from the impact energy of the sprayed feedstock, the presence of an oxide layer plays a crucial role in determining the adhesion strength of the coating. The oxide layer affects the hardness and structure of the powder, and its thickness influences the critical speed required to rupture it and establish a bond between the powders and the substrate. The presence of residual oxide within the coating is detrimental to both powder deposition and the performance of the coating itself (as depicted in Figure 9a). When using dendritic powder in cold spraying, the deformation of the powder is minimal, resulting in limited erosion of the substrate and poor coating adhesion. This inadequate adhesion can be attributed to the irregular morphology of dendritic powder, which disperses the impact force. Consequently, when spraying dendritic powders with an oxide layer, a higher impact force is necessary to break the powder boundaries. Failure to achieve sufficient plastic deformation leads to incomplete overlap of the powders and the formation of numerous pores, with the oxide layer persisting along the powder boundaries (as illustrated in Figure 9b).

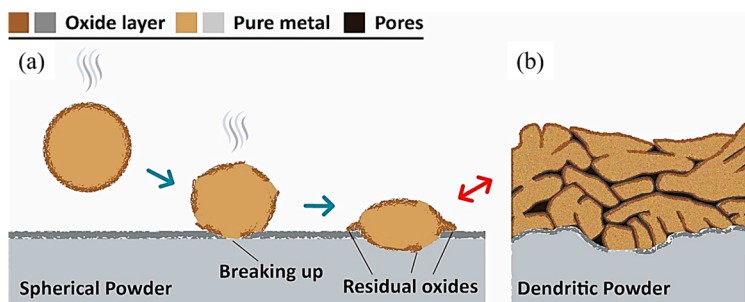

**Figure 9.** Schematic drawing illustrating the coating formation from (**a**) spherical and (**b**) dendritic powders with oxygen layer.

## 4. Conclusions

This article investigates cold-sprayed coatings made from gas-atomized and electrolytic powders. It analyzes the microstructure and adhesion strength of the coatings and proposes a formation mechanism for the cold spray process. Coatings formed with gas-atomized powders had a lamellar structure, while those made with electrolytic powders had a dense stacking pattern. The coatings from electrolytic powders had higher density and a smoother interface with the substrate. Gas-atomized powders had fewer noticeable pores. Adhesion strength was higher for gas-atomized powders, spherical powders, and larger particle sizes. The article suggests a formation mechanism where kinetic energy converted to heat energy softens the substrate surface, allowing particles to deform and adhere, forming a crater-like shape. The presence and thickness of an oxide layer affected adhesion strength. Incomplete deformation of dendritic powders led to poor adhesion and the formation of pores at the interface. This article examines the characteristics of cold-sprayed coatings using different powders. It discusses the effects of powder morphology, particle size, and oxide layer on coating properties. The proposed formation mechanism enhances our understanding of the cold spray process. The findings have practical implications for optimizing cold spray parameters and improving coating quality for diverse applications.

**Author Contributions:** Validation, K.-C.C.; investigation, C.-H.W.; resources, C.-W.T.; data curation, K.H.; writing—original draft preparation, J.-T.L. and S.-F.C.; writing—review and editing, S.-H.C. All authors have read and agreed to the published version of the manuscript.

**Funding:** This work was financially supported by the "High Entropy Materials Center" from The Featured Areas Research Center Program within the framework of the Higher Education Sprout Project by the Ministry of Education (MOE) and the Project NSTC 111-2634-F-007-008 and 111-2221-E-A49-195—by the National Science and Technology Council (NSTC) in Taiwan.

**Institutional Review Board Statement:** Not applicable.

**Informed Consent Statement:** Not applicable.

**Data Availability Statement:** Not applicable.

**Conflicts of Interest:** The authors declare no conflict of interest.

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
