# Peer review of "Influence of Feedstock in the Formation Mechanism of Cold-Sprayed Copper Coatings"

_coatings, doi:10.3390/coatings13061065_

Round 1

Reviewer 1 Report

The present work entitled "Influence of feedstock in the formation mechanism of cold-sprayed copper coatings" considers the influence of various copper powders on the properties of coatings obtained by cold spraying.

 This topic is interesting and worthy to be investigated.

The methodological part is described in detail.

 Nevertheless, there are several comments to the authors:

 1. There are no descriptions of such deposition characteristics as the speed of the nozzle movement and the percentage of track overlap during deposition.

2. The structure of the resulting coatings has not been studied in sufficient detail. There are no microstructures with higher magnification. There are no results of the analysis of the chemical composition of the coatings.

3. It is not clear how the density and porosity of the coatings were compared.

4. Contradictions in the abstract and conclusions. First, it was noted that gas-atomized powder coatings have pores, while electrolytic powder coatings are denser (lines 18-20 and 320-323). Then it is said that coatings made of dendritic powders are prone to the formation of pores (lines 28 and 335-337).

5. Line 110: 105 K/s

6. Line 144: 6061 aluminum alloy substrates were selected as the substrate

7. Line 151: What is reciprocating spraying?

8. Lines 218-241: This is more of a methodical part than results.

9. Line 355: Reference “Author 1, A.B.; Author 2, C.D. Title of the article. Abbreviated Journal Name Year, Volume, page range.”

Broken References numbering

10. Conclusions need to be written more concisely and inconsistencies need to be corrected.

The paper with opportune modifications can be considered for publication

Author Response

Reviewer #1: The present work entitled "Influence of feedstock in the formation mechanism of cold-sprayed copper coatings" considers the influence of various copper powders on the properties of coatings obtained by cold spraying. This topic is interesting and worthy to be investigated. The methodological part is described in detail.

Nevertheless, there are several comments to the authors:

  1. There are no descriptions of such deposition characteristics as the speed of the nozzle movement and the percentage of track overlap during deposition.

Response: Thanks to Reviewer’s kindly suggestion. More parameters of cold spraying process were included in Table 2.

  1. The structure of the resulting coatings has not been studied in sufficient detail. There are no microstructures with higher magnification. There are no results of the analysis of the chemical composition of the coatings.

Response: Thanks to Reviewer’s kindly suggestion. We have investigated the chemical compositions of all powders; however, as the results of XRD analysis, no obvious difference was noticed. Thus, only XRD spectra was used to show the uniformity and high purity of Cu materials. The example of Cu powders and coatings are shown as below.

Figure 7 presents the microstructure of sprayed samples, which might be sufficient to show the mechanism of coating formation.

Figure A. Element mapping of cold-sprayed sample and the EDS analysis of electrolysis Cu powders.

  1. It is not clear how the density and porosity of the coatings were compared.

Response: Thanks to Reviewer’s kindly suggestion. In this study, we focused on the formation and stacking mechanism of cold-sprayed coatings associated with the feedstock. The comparison of density was described by the observation in microstructure.

  1. Contradictions in the abstract and conclusions. First, it was noted that gas-atomized powder coatings have pores, while electrolytic powder coatings are denser (lines 18-20 and 320-323). Then it is said that coatings made of dendritic powders are prone to the formation of pores (lines 28 and 335-337).

Response: The first sentence is used to describe the stacking between powders, and the second one present the bonding situation at the interface. In order to make the statement clearer, more information was added in the text.

  1. Line 110: 105 K/s

Response: It was corrected as 105 K/s.

  1. Line 144: 6061 aluminum alloy substrates were selected as the substrate

Response: It was corrected as “6061 aluminum alloy was selected as the substrate

  1. Line 151: What is reciprocating spraying?

Response: It means the spraying route is to move backward and forward in a straight line on the substrate till the entire surface was covered.

  1. Lines 218-241: This is more of a methodical part than results.
  2. Line 355: Reference “Author 1, A.B.; Author 2, C.D. Title of the article. Abbreviated Journal Name Year, Volume, page range.” Broken References numbering

Response: It was removed.

  1. Conclusions need to be written more concisely and inconsistencies need to be corrected.

Response: Thanks to Reviewer’s kindly suggestion. The conclusion was rewording for a better readability.

Reviewer 2 Report

       The manuscript describes the area of research with good analysis. However, it needs certain modifications to be implemented for better readability and understanding of the work. The following comments are suggested:

1. The abstract should explain the aim and significance of the work accomplished including the main findings of work.

2.    The introduction section needs to include more number of recent literature indicating the research gaps being focussed in the manuscript.

3.   What is the significance of using cold sprayed coatings and why is it constructed by gas-atomized and electrolytic powders?

4.       How the parameters were selected for the experimentation?

5.    Kindly ensure that all the figures and tables are cited near their actual position in the manuscript with important information required including proper justifications.

6.       The results and discussion section requires a more detailed explanation of the results (graphs) validated from the literature.

7.     Explain the novelty of the work presented with respect to other recent researches in the field.

8.     Rewrite the conclusion section for explaining the significance, aim and introduction of the work in addition to the results of the study.

9.       Explain the relevant future work for the study.

Author Response

Reviewer #2: The manuscript describes the area of research with good analysis. However, it needs certain modifications to be implemented for better readability and understanding of the work. The following comments are suggested:

  1. The abstract should explain the aim and significance of the work accomplished including the main findings of work.

Response: Thanks to Reviewer’s kindly suggestion. The abstract was rewording for a better readability.

  1. The introduction section needs to include more number of recent literature indicating the research gaps being focused in the manuscript.

Response: Thanks to Reviewer’s kindly suggestion. The section was checked and modified for a better readability.

  1. What is the significance of using cold sprayed coatings and why is it constructed by gas-atomized and electrolytic powders?

Response: This dry deposition process can provide a relatively clean approach without wastewater, and can apply to almost all kind of surfaces with a suitable roughness.

  1. How the parameters were selected for the experimentation?

Response: Our industry partner suggested this spraying parameter used in mass-produced products.

  1. Kindly ensure that all the figures and tables are cited near their actual position in the manuscript with important information required including proper justifications.

Response: Thanks to Reviewer’s kindly suggestion. All the figures and tables in this article are re-checked and necessary.

  1. The results and discussion section requires a more detailed explanation of the results (graphs) validated from the literature.

Response: Thanks to Reviewer’s kindly suggestion. The section was checked and modified for a better readability.

  1. Explain the novelty of the work presented with respect to other recent researches in the field.

Response: Thanks to Reviewer’s kindly suggestion. The motivation was addressed in the introduction.

  1. Rewrite the conclusion section for explaining the significance, aim and introduction of the work in addition to the results of the study.

Response: The conclusion and introduction parts were modified for a better readability.

  1. Explain the relevant future work for the study.

Response: This work extended from the development in the cooling devices of battery system for electric vehicles. 
